

# Supersymmetric polynomials and algebro-combinatorial duality

Dmitry Galakhov[2,3,4*], Alexei Morozov[1,2,3,4†] and Nikita Tselousov[1,2,3,4‡]

**1** MIPT, 141701, Dolgoprudny, Russia
**2** NRC "Kurchatov Institute", 123182, Moscow, Russia
**3** IITP RAS, 127051, Moscow, Russia
**4** ITEP, Moscow, Russia

⋆ galakhov@itep.ru , † morozov@itep.ru , ‡ tselousov.ns@phystech.edu

## Abstract

In this note we develop a systematic combinatorial definition for constructed earlier *supersymmetric polynomial* families. These polynomial families generalize canonical Schur, Jack and Macdonald families so that the new polynomials depend on odd Grassmann variables as well. Members of these families are labeled by respective modifications of Young diagrams. We show that the super-Macdonald polynomials form a representation of a super-algebra analog $\mathsf{T}(\widehat{\mathfrak{gl}}_{1|1})$ of Ding-Iohara-Miki (quantum toroidal) algebra, emerging as a BPS algebra of D-branes on a conifold. A supersymmetric modification for Young tableaux and Kostka numbers are also discussed.



## Contents

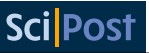

# 1   Introduction

This paper is a satellite paper to [1]. Here we develop a systematic, unified and, most importantly, combinatorial approach to constructing super-Schur, super-Jack and super-Macdonald polynomial families defined earlier (see also [2]).

As usual the prefix "super" implies that one mixes usual commuting even variables with new odd anti-commuting Grassmann variables acquiring extra phase $(-1)$ under permutation. And now (anti-)symmet- rization in polynomials of these variables includes new variables as well.

Here we choose our approach as the basic one so it reflects and generalizes naturally approaches to ordinary Schur, Jack and Macdonald symmetric functions in canonical textbooks [3] on this subject. Members of supersymmetric polynomial families are labeled by super-partitions, or super-Young diagrams, that generalize the notion of ordinary linear partitions and inherit many of its properties. The construction is based on an extension of a notion of basic symmetrized monomials now in a ring of both even and odd variables. Then the family of polynomials is constructed as a set of orthogonal polynomials having an upper-triangular decomposition over basic monomials with respect to an ordered set of super-Young diagrams and to a respective norm, Schur, Jack or Macdonald.

In fact, there are two inequivalent orders on (super-)Young diagrams. And an equivalence (Kerov self-duality [4]) of upper-triangular mutual re-expansions of polynomial families, say, (super-)Macdonald polynomials over (super-)Schur polynomials, was the guiding principle to define new super-Macdonald polynomials in [1]. Here we confirm that our supersymmetric polynomial families are insensitive with respect to the order choice in expansions with respect to basic monomials as well.

However we should stress that we are not aware what is the fundamental principle to implement triangular decompositions yet. Here we simply start with a theory of symmetric functions just assuming it is more fundamental than the theory of Schur/Jack/Macdonald polynomials. A combinatorial approach to supersymmetric polynomials based on orthogonality properties was also developed in a series of papers [5–10]. In the context of CKP hierarchy polynomials similar to what we call super-Schur polynomials were introduced in [11].

Furthermore, having constructed families of supersymmetric polynomials we observe they form semi-Fock representations of certain algebras: super-Macdonald polynomials for super-symmetrized Ding-Iohara-Miki (quantum toroidal) algebra $\mathsf{T}(\widehat{\mathfrak{gl}}_{1|1})$, whereas super-Jack and super-Schur polynomials are representations for affine super-Yangian $\mathsf{Y}(\widehat{\mathfrak{gl}}_{1|1})$. This observation stands in a search line [2, 12–17] for bosonized representations of quiver BPS algebras associated to D-brane systems on toric Calabi-Yau three-folds [18–21] (see also reviews in [22–25]). In this particular case $\mathsf{T}(\widehat{\mathfrak{gl}}_{1|1})$ is a trigonometric deformation [26, 27] of affine quiver Yangian on a smooth toric variety representing stable perverse coherent sheaves on the blowup of a projective surface [28, 29].

Nevertheless we find this convergence of seemingly different combinatorial and algebraic approaches to constructing supersymmetric polynomials rather surprising and call this phenomenon an *algebro-combinatorial duality*.

In general there are four ways to define a family of (super)symmetric polynomials wherein family members are labeled by (super-)Young diagrams:

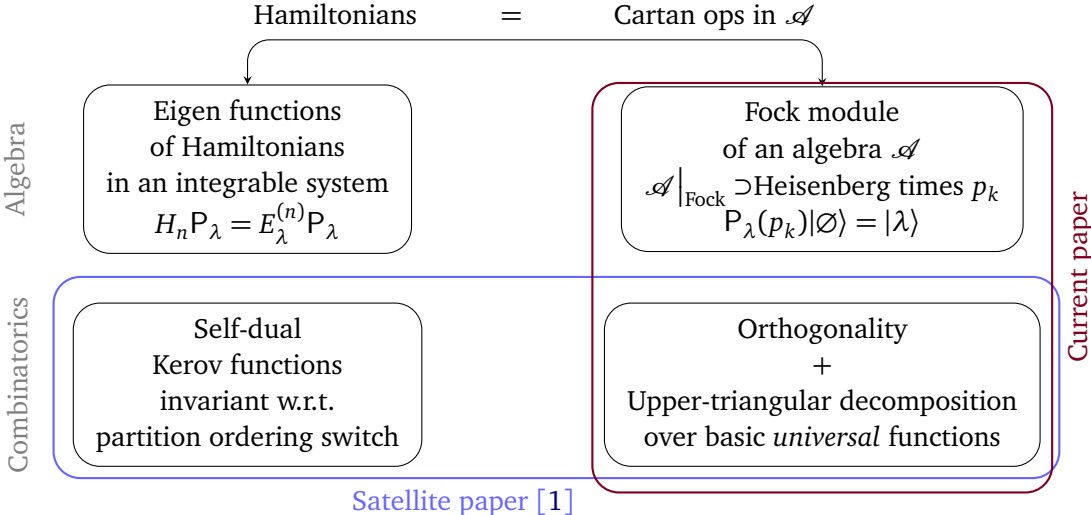

In the current note we cover only a relation of two approaches. However it becomes clear from this scheme that supersymmetric polynomials could be defined as eigen function of a system of commuting Hamiltonians generalizing Hamiltonians in Calogero-like integrable systems [30–37]. In particular, commuting Cartan generators in $\mathsf{T}(\widehat{\mathfrak{gl}}_{1|1})$ in the free-field representation would become an infinite tower of commuting operators in time variables. Some hints on the form of these new super-Macdonald Hamiltonians are given in [1] and in App. A. An explicit construction will be presented elsewhere.

Finally, but not least, we should mention that supersymmetric polynomials extend from the ordinary case an intriguing combinatorial interpretation for coefficients in the upper-triangular decomposition as sums over Young tableaux. In this note we were able to extend the notion of the Young tableaux to the super-Young tableaux and the combinatorial interpretation as well. However unlike the ordinary case in the super-case appropriate weights for super-Young tableaux in the decomposition coefficients do not factorize in general. We call these weights *super-Kostka amplitudes* after analogous Kostka numbers in the case of Schur functions.

This paper is organized as follows. In Sec. 2 we remind the construction of super-partitions, super-Young diagrams and introduce super-Young tableaux. In Sec. 3 we present our combinatorial construction for supersymmetric polynomials. Sec. 4 is devoted to a representation of $\mathsf{T}(\widehat{\mathfrak{gl}}_{1|1})$ on super-Macdonald polynomials. In Sec. 5 we describe an origin for super-Young tableaux in a theory of supersymmetric polynomials and attempt to derive super-Kostka amplitudes.

## 2 Super-partitions

### 2.1 Super-partitions and super-Young diagrams

In this paper we will extensively exploit the field of half-integer numbers $\mathbb{Z}/2$. We will call ordinary integer numbers in this field *even* and non-integer numbers *odd*. This imposes on half-integer numbers $j \in \mathbb{Z}/2$ a natural $\mathbb{Z}_2$-grading:[1]

$$\mathfrak{f}(j) := 2(j - \lfloor j \rfloor). \tag{1}$$

---

[1]Here we use a notation $\lfloor x \rfloor$ (resp. $\lceil x \rceil$) for the floor (resp. the ceiling) function of $x$.

We call a *super-partition* $\lambda$ of half-integer non-negative number $n \in \mathbb{Z}_{\geq 0}/2$ an ordered sequence of half-integer numbers $\lambda_i$ satisfying inequalities:

$$\lambda_1 \geq \lambda_2 \geq \lambda_3 \geq \ldots \geq \lambda_k \geq 0, \qquad |\lambda| = \sum_{i=1}^{k} \lambda_i = n, \tag{2}$$

such that for odd numbers $\mathfrak{f}(\lambda_i) = 1$ in this sequence inequalities are strict $\lambda_{i-1} > \lambda_i > \lambda_{i+1}$. By $\ell(\lambda)$ we will denote the number of non-zero elements in $\lambda$.

For a partition $\lambda$ we distinguish its even and odd parts and denote them as $\lambda^+$ and $\lambda^-$ respectively. For example,

$$\lambda = \{{}^{11}/_2, 3, {}^5/_2, 1, 1, {}^1/_2\}, \qquad \lambda^+ = \{3, 1, 1\}, \qquad \lambda^- = \{{}^{11}/_2, {}^5/_2, {}^1/_2\}. \tag{3}$$

Ordinary linear partitions were identified naturally with Young diagrams. Analogously we identify super-partitions with *super-Young diagrams*. Since this is a one-to-one correspondence partitions and diagrams will not be distinguished in notations. A super Young diagram $\lambda$ is a collection of full tiles $\square$ and upper half-tiles $\triangleright$ filling out a lower right quadrant corner of a plane so that each tile is supported on its top and its left sides either by another tile or the corner wall. Identification between diagrams and partitions goes as follows. To each $\lambda_i \in \lambda$ we identify a *column* of tiles of length $\lceil \lambda_i \rceil$, and the most bottom tile is a complete tile $\square$ if $\mathfrak{f}(\lambda_i) = 0$, or a half-tile $\triangleright$ if $\mathfrak{f}(\lambda_i) = 1$. For example,

$$\{6, {}^{11}/_2, {}^9/_2, 4, 3, 3, {}^5/_2, 1, 1\} = \quad . \tag{4}$$

In what follows we will also divide a complete tile $\square$ in a combination of an upper half-tile $\triangleright$ and a lower half-tile $\triangle$.

The generating function for the number of super-partitions reads:

$$\sum_{\lambda} q^{2|\lambda|} = \prod_{k=1}^{\infty} \frac{1 + q^{2k-1}}{1 - q^{2k}}. \tag{5}$$

Let us enumerate them for the first few levels:

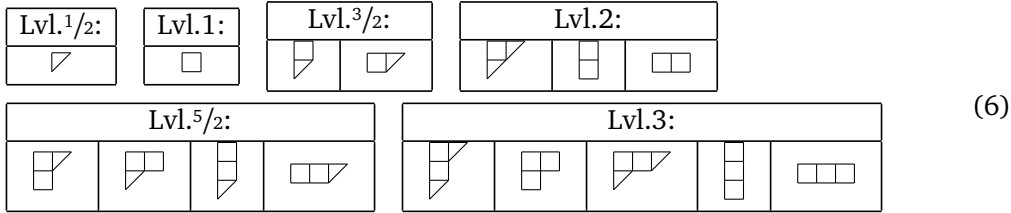

$$\tag{6}$$

Apparently, the set of super-partitions (resp. super-Young diagrams) include a set of ordinary linear partitions (resp. ordinary Young diagrams) as a subset.

In what follows we will also need a notion of transposition on super-partitions. Transposition acts naturally on super-Young diagrams reflecting it with respect to the diagonal ray passing thorough it. For example,

$$\{4, {}^5/_2, 1, {}^1/_2\} = \quad , \qquad \{4, {}^5/_2, 1, {}^1/_2\}^T = \{{}^7/_2, 2, {}^3/_2, 1\} = \quad . \tag{7}$$

Finally, we note that partitions at the same level form an *ordered* set. We say that $\lambda > \mu$ if

$$|\lambda| = |\mu|, \quad \text{and} \quad \exists k : \lambda_i = \mu_i, \qquad \text{for} \quad i < k, \quad \text{and} \quad \lambda_k > \mu_k. \tag{8}$$

For example,

$$\text{(diagrams)} \tag{9}$$

## 2.2 Super-Young tableaux

Let us make a brief reminder on a Young tableau construction for purely even diagrams $\ell(\lambda^-) = 0$ following [3],[2] then we switch to our suggestions to generalize this definition to super-Young diagrams.

A Young tableau of shape $\lambda$ and weight $\mu$ is a collection of numbers: $\mu_1$ of 1, $\mu_2$ of 2, $\mu_3$ of 3, and so on, arranged in cells of Young diagram $\lambda$ in such a way that numbers do not decrease as we move between cells inside a column and strictly increase as we move inside a row.

We will denote a set of all (super-)Young tableaux of diagram $\lambda$ with weight $\mu$ as $T(\lambda, \mu)$. So, for example, for purely even diagrams we have:

$$T\left(\text{\ },\ \text{\ }\right) = \left\{ \text{\ }, \ \text{\ }, \ \text{\ }, \ \text{\ } \right\}. \tag{10}$$

Let us stress here that the second argument $\mu$ in the set valued function $T(\lambda, \mu)$ is *not necessarily* a Young diagram (resp. a super-Young diagram in what follows.) It could be any sequence of non-negative integer (resp. half-integer) numbers. However in triangular decompositions below the second argument $\mu$ is not generic – it is always a (super-)Young diagram. This is at least amusing, anyhow, in what follows we will work mostly with the cases when both $\lambda$ and $\mu$ are (super-)Young diagrams.

It is apparent from this definition that if $\mu$ is a Young diagram and $\mu > \lambda$ then for such diagrams $T(\lambda, \mu) = \varnothing$ as otherwise we will have to arrange equal numbers in a row that is forbidden by our conventions:

$$T\left(\text{\ },\ \text{\ }\right) = \left\{ \text{\ } \right\} = \varnothing. \tag{11}$$

We are not aware if there is a simple depiction for super-Young tableaux in the fashion of the ordinary Young tableaux as Young diagrams with constrained numbers in cells. The reason is that in the super-case we will have to fill out with numbers half-tiles rather than complete cells, and it becomes problematic to work out a concise rule for such a filling. Therefore we generalize an alternative definition of Young tableau as a sequence $\tau^i$ of embedded diagrams. One constructs elements $\tau^i$ of this sequence as diagrams of cells of a tableau with inscribed numbers $k \le i$, for example,

$$\text{(diagram)} = \varnothing \subset \text{\ } \subset \text{\ } \subset \text{\ } \subset \text{\ } \subset \text{\ } \subset \text{\ }. \tag{12}$$

Then we introduce a definition of a vertical strip compatible with Young and super-Young diagrams. For a pair of partitions $\lambda$ and $\mu$ we say that $\lambda/\mu$ is a *vertical strip* of length $n \in \mathbb{Z}/2$ if $|\lambda| - |\mu| = n$ and in each row of a diagram complement $\lambda/\mu$ there is at maximum one upper

---

[2]In comparison to [1] and [3] in this paper we interchanged diagram rows and columns in our conventions.

half-tile $\triangledown$ and one lower half-tile $\triangle$. In other words, we could reformulate this constraint in terms of transposed partitions:

$$|\lambda| - |\mu| = n, \qquad 0 \le \lambda_i^T - \mu_i^T \le 1, \quad \forall i. \tag{13}$$

If $\lambda/\mu$ is a vertical strip of length $n$ we will mark such a situation as $|\lambda/\mu| = n$.

We call a *super-Young tableau* $\tau$ of diagram $\lambda$ and of weight $\nu$ a sequence of super-Young diagrams $\tau^i$:

$$\varnothing = \tau^0 \subset \tau^1 \subset \tau^2 \subset \ldots \subset \tau^p = \lambda, \tag{14}$$

such that $\tau^i/\tau^{i-1}$ is a vertical strip of length $\nu_i$.

For example, we could have constructed the following tableau of shape $\lambda$ and weight $\mu$:

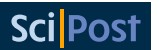
$$\begin{aligned} \lambda &= \{{}^7\!/_2, 3, {}^3\!/_2, {}^1\!/_2\}, \\ \mu &= \{{}^5\!/_2, 2, 2, {}^3\!/_2, {}^1\!/_2\}, \end{aligned} \tag{15}$$

$$\varnothing \subset \{{}^5\!/_2\} \subset \{3, {}^3\!/_2\} \subset \{3, {}^5\!/_2, 1\} \subset \{{}^7\!/_2, 3, 1, {}^1\!/_2\} \subset \{{}^7\!/_2, 3, {}^3\!/_2, {}^1\!/_2\},$$

where we used the same color code for tiles and weight elements as they contribute to.

## 3 Supersymmetric polynomials

### 3.1 Simple bases in supersymmetric polynomial ring

Canonically Schur, Jack and Macdonald polynomials are defined as distinguished bases in a ring $\mathcal{R}$ of symmetric polynomials in commuting variables $x_i$. Another simple canonical basis in this ring of $N$ variables is spanned by symmetric monomials:

$$m_\lambda(x_1, x_2, \ldots, x_N) = \sum_{\alpha \in S_N \cdot \lambda} x_1^{\alpha_1} x_2^{\alpha_2} \ldots x_N^{\alpha_N}, \tag{16}$$

where $S_N$ are various permutations of elements in a Young diagram $\lambda$.

The symmetric polynomial ring is generated by simply symmetrized powers:

$$p_k = m_{\{k\}} = \sum_i x_i^k. \tag{17}$$

Then we could have chosen another natural basis in $\mathcal{R}$:

$$P_\lambda := \prod_i p_{\lambda_i}, \tag{18}$$

where $\lambda$ are Young diagrams.

In usual discussions around integrable systems variables $x_i$ are usually called Miwa variables, identification (17) a Miwa transform, and variables $p_k$ are called times, as they serve as time parameters conjugated to Hamiltonians of a Calogero system [30–32].

To promote the discussion to the super-case let us also consider Grassmann Miwa variables $\xi_i$ that mutually anti-commute $\xi_i \xi_j + \xi_j \xi_i = 0$ and commute with variables $x_i$. Now we are able to consider a ring of supersymmetric polynomials $\mathcal{S}$, where the symmetrization procedure permutes variables $\xi_i, \xi_j$ with the $(-1)$-sign.

To simplify notations in the super-case let us define super-time variables generating ring $\mathcal{S}$ with half-integer indices $j \in \mathbb{Z}_{\ge 0}/2$:

$$p_j := \begin{cases} \sum_i x_i^j, & \text{if } \mathfrak{f}(j) = 0, \\ \sum_i x_i^{j-\frac{1}{2}} \xi_i, & \text{if } \mathfrak{f}(j) = 1. \end{cases} \tag{19}$$

Variables $p_j$ super-commute:

$$\left[p_j, p_{j'}\right\} := p_j p_{j'} - (-1)^{\mathfrak{f}(j)\mathfrak{f}(j')} p_{j'} p_j = 0, \tag{20}$$

and together with respective derivatives $p_{-j} := \partial_{p_j}$ form a super-Heisenberg algebra:

$$\left[p_j, p_{j'}\right\} = \delta_{j+j',0}, \qquad j, j' \in (\mathbb{Z}/2 \setminus \{0\}). \tag{21}$$

Again we could choose a basis for $\mathscr{S}$ made of monomials of super-time variables:

$$P_\lambda = \prod_i p_{\lambda_i}, \tag{22}$$

where $\lambda$ are super-Young diagrams, and we always assume that multipliers appear in the r.h.s. in the order as they appear in the partition.

A basis for $\mathscr{S}$ in terms of symmetric monomials is also available in the super-case:

$$m_\lambda(x_1, x_2, \ldots, x_N, \xi_1, \ldots, \xi_N) = \sum_{\alpha \in S_N \cdot \lambda} \mathbf{x}_1^{\alpha_1} \mathbf{x}_2^{\alpha_2} \ldots \mathbf{x}_N^{\alpha_N}, \tag{23}$$

where $\lambda$ are again super-Young diagrams,

$$\mathbf{x}_i^k := \begin{cases} x_i^k, & \mathfrak{f}(k) = 0, \\ x_i^{k-\frac{1}{2}} \xi_i, & \mathfrak{f}(k) = 1. \end{cases} \tag{24}$$

Since $P_\lambda$ and $m_\lambda$ are two alternative bases in the same $\mathscr{S}$ we could always decompose one basis elements over the others. However it is not very convenient to exploit the definitions in terms of Miwa variables to construct this decomposition.

Instead let us state a simple multiplication rule allowing to reconstruct this decomposition easily.

It seems to be easier to formulate this rule in terms of monomials $M_\lambda$ normalized differently:

$$M_\lambda := \frac{z_\lambda}{\prod\limits_{x \in \lambda^+} x} m_\lambda, \tag{25}$$

where $z_\lambda = z_{\lambda^+}$, and for purely even partition $\lambda$ $z$-norm is defined in a canonical way [3, 38]:

$$\text{if} \quad \lambda = (1^{n_1}, 2^{n_2}, 3^{n_3}, \ldots), \quad \text{then} \quad z_\lambda = \prod_k k^{n_k} n_k!. \tag{26}$$

We start with a boundary condition:

$$M_{\{j\}} := P_{\{j\}} = p_j, \qquad j \in \mathbb{Z}/2. \tag{27}$$

Then multiplication by a column monomial on the right could be expanded over monomials again with simple coefficients:

$$M_\lambda M_{\{x\}} = \sum_{k=1}^{\ell(\lambda)+1} (-1)^{\mathfrak{f}(x) \sum\limits_{i=k}^{\ell(\lambda)} \mathfrak{f}(\lambda_i)} \left(1 - \delta_{\mathfrak{f}(x),1} \delta_{\mathfrak{f}(\lambda_k),1}\right) M_{:\lambda + x e_k:}, \tag{28}$$

where $e_k$ is a unit vector with a unit entry at the $k^{\text{th}}$ position. This seemingly complicated expression is in practice a fairly simple combinatorial action. Multiplication by $\{x\}$ is an operator that tries to add $x$ to each element of $\lambda$ to get a new partition. However depending on parity of $x$ it is either bosonic (even) or fermionic (odd) $p_x$, so it (anti-)commutes with

other elements of $\lambda$ depending on their parity. This operator can not add a half-integer to a half-integer, this explains the second term in the r.h.s. of (28). Finally the new partition should be normal ordered. When we permute fermions (half-integers), this action contributes with a minus sign. And when two equal half-integers are encountered, this contributes with 0.

For example,

$$
\begin{aligned}
M_{\{1/2\}} M_{\{1\}} &= M_{\{1,1/2\}} + M_{\{3/2\}}\,, \\
M_{\{3/2\}} M_{\{1/2\}} &= M_{\{3/2,1/2\}} = -M_{\{1/2\}} M_{\{3/2\}}\,.
\end{aligned}
\tag{29}
$$

Rule (28) allows one to construct a decomposition of $M_\lambda$ and $m_\lambda$ over $P_\lambda$ for all $\lambda$.

Let us present this supersymmetric polynomial basis re-decomposition for the first few levels:

- Level $1/2$:

$$
m_{\{1/2\}} = P_{\{1/2\}}\,.
\tag{30}
$$

- Level 1:

$$
m_{\{1\}} = P_{\{1\}}\,.
\tag{31}
$$

- Level $3/2$:

$$
m_{\{3/2\}} = P_{\{3/2\}}\,, \qquad m_{\{1,1/2\}} = P_{\{1,1/2\}} - P_{\{3/2\}}\,.
\tag{32}
$$

- Level 2:

$$
m_{\{2\}} = P_{\{2\}}\,, \qquad m_{\{3/2,1/2\}} = P_{\{3/2,1/2\}}\,, \qquad m_{\{1,1\}} = \frac{1}{2}P_{\{1,1\}} - \frac{1}{2}P_{\{2\}}\,.
\tag{33}
$$

- Level $5/2$:

$$
\begin{aligned}
m_{\{5/2\}} &= P_{\{5/2\}}\,, & m_{\{2,1/2\}} &= P_{\{2,1/2\}} - P_{\{5/2\}}\,, \\
m_{\{3/2,1\}} &= P_{\{3/2,1\}} - P_{\{5/2\}}\,, & m_{\{1,1,1/2\}} &= P_{\{5/2\}} - P_{\{3/2,1\}} - \frac{1}{2}P_{\{2,1/2\}} + \frac{1}{2}P_{\{1,1,1/2\}}\,.
\end{aligned}
\tag{34}
$$

- Level 3:

$$
\begin{aligned}
m_{\{3\}} &= P_{\{3\}}\,, & m_{\{5/2,1/2\}} &= P_{\{5/2,1/2\}}\,, & m_{\{2,1\}} &= -P_{\{3\}} + P_{\{2,1\}}\,, \\
m_{\{3/2,1,1/2\}} &= -P_{\{5/2,1/2\}} + P_{\{3/2,1,1/2\}}\,, & m_{\{1,1,1\}} &= \frac{1}{3}P_{\{3\}} - \frac{1}{2}P_{\{2,1\}} + \frac{1}{6}P_{\{1,1,1\}}\,.
\end{aligned}
\tag{35}
$$

## 3.2 (Super) Schur, Jack and Macdonald polynomials

One way to define Schur, Jack and Macdonald polynomial bases in the ring of symmetric polynomials $\mathscr{R}$ is as orthogonal polynomials with respect to a specific norm that has a triangular decomposition over basis $m_\lambda$ in $\mathscr{R}$ [3,38]. For a generic choice of norm respective polynomial families are known as Kerov polynomials [4].

Let us define a basis of polynomials $\mathsf{P}_\lambda$ labeled by super-Young diagrams $\lambda$ in the supersymmetric polynomial ring $\mathscr{S}$ following two rules:[3]

---

[3]For computational purposes the following recurrent relation is also useful:

$$
\mathsf{P}_\lambda = m_\lambda - \sum_{\mu < \lambda} \frac{\langle m_\lambda, \mathsf{P}_\mu \rangle}{\langle \mathsf{P}_\mu, \mathsf{P}_\mu \rangle} \mathsf{P}_\mu\,.
$$

$$\text{1) Upper-triangular decomposition:} \quad \mathsf{P}_\lambda = m_\lambda + \sum_{\mu < \lambda} K_{\lambda\mu}\, m_\mu\,,$$

$$\text{2) Orthogonality:} \quad \langle \mathsf{P}_\lambda, \mathsf{P}_\mu \rangle = 0\,, \quad \text{unless} \quad \lambda = \mu\,. \tag{36}$$

These polynomials generalize canonical Schur, Jack and Macdonald polynomials to the super-Young diagrams, therefore we call them respectively depending on the norm choice for super-time variables:

| Name | P | Norm |
|---|---|---|
| super-Schur | S | $\langle P_\lambda, P_\mu \rangle_{\mathrm{S}} = \delta_{\lambda\mu} z_\lambda$ |
| super-Jack | J | $\langle P_\lambda, P_\mu \rangle_{\mathrm{J}} = \delta_{\lambda\mu} z_\lambda \beta^{\ell(\lambda^+)}$ |
| super-Macdonald | M | $\langle P_\lambda, P_\mu \rangle_{\mathrm{M}} = \delta_{\lambda\mu} z_\lambda \times \prod_{x \in \lambda^+} \dfrac{1-q^{2x}}{1-t^{2x}} \times \prod_{y \in \lambda^-} q^{2y}$ |

$(37)$

There is a certain freedom [4] in the choice of the canonical norms related to rescaling/re-defining time variables. In this paper we work with conventions (37).

In practice, we could have chosen another canonical order. Let us call the order defined in (8) a column order $\texttt{<=:}\prec_c$. We could have taken a row order $\prec_r$ [1] defined as:

$$\lambda \prec_r \mu \quad \Longleftrightarrow \quad \lambda^T \succ_c \mu^T\,. \tag{38}$$

Despite at low levels these orders coincide, at higher orders they start to differ substantially. In particular, they differ at level 8 for purely even diagrams, and at level $^5/_2$ these orders start to mix an even diagram with an odd diagram, whereas at level $^9/_2$ they mix two odd diagrams, and so on.[4] For a generic norm choice resulting Kerov polynomial families will be different for two orders.

> However, as it was observed in [1] the super-Schur/Jack/Macdonald polynomials *coincide* for two orders $\prec_c$ and $\prec_r$.

In fact these "miraculous" coincidence was the guiding principle for the super-Macdonald norm choice (37).

Here let us list super-Macdonald polynomials for the first few levels:

- Level $^1/_2$:
$$\mathsf{M}_{\{^1/_2\}} = P_{\{^1/_2\}}\,. \tag{39}$$

- Level 1:
$$\mathsf{M}_{\{1\}} = P_{\{1\}}\,. \tag{40}$$

- Level $^3/_2$:

$$\mathsf{M}_{\{^3/_2\}} = \frac{q^2\left(t^2 - 1\right)}{q^2 t^2 - 1} P_{\{1, ^1/_2\}} + \frac{\left(q^2 - 1\right)}{q^2 t^2 - 1} P_{\{^3/_2\}}\,, \qquad \mathsf{M}_{\{1, ^1/_2\}} = P_{\{1, ^1/_2\}} - P_{\{^3/_2\}}\,. \tag{41}$$

---

[4]Compared to [1] we do not split here sequences of super-partitions at even levels in purely even partitions and all the others. Respective sequences turn out to be orthogonal $K_{\lambda\mu} = 0$ on their own.

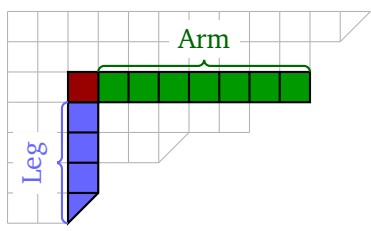

Figure 1: Hook, arm and leg in a super-Young diagram.

- Level 2:

$$M_{\{2\}} = \frac{\left(q^2+1\right)\left(t^2-1\right)}{2q^2t^2-2}P_{\{1,1\}} + \frac{\left(q^2-1\right)\left(t^2+1\right)}{2q^2t^2-2}P_{\{2\}}\,,$$
$$M_{\{3/2,1/2\}} = P_{\{3/2,1/2\}}\,,$$
$$M_{\{1,1\}} = \frac{1}{2}P_{\{1,1\}} - \frac{1}{2}P_{\{2\}}\,. \tag{42}$$

For purely even diagrams $\ell(\lambda^-) = 0$ super-Macdonald polynomials *coincide* with canonically defined Macdonald polynomials for Young diagrams [3].

In what follows we will mostly work with super-Macdonald polynomials. Since the norm definition for them is the most sophisticated, the properties of the super-Schur and super-Jack polynomials simply follow.

Scalar norm for the super-Macdonald polynomials could be computed explicitly in terms of hook functions (see Fig. 1):

$$\left\langle M_\lambda, M_\mu \right\rangle_{\mathrm M} = \delta_{\mu\nu} \times \mathcal{N}_\lambda\,,$$
$$\mathcal{N}_\lambda = q^{\ell(\lambda^-)} \times \prod_{\square \in \lambda} \upsilon_\lambda(\square)\,, \tag{43}$$

where the product runs over complete tiles in diagram $\lambda$. Hook functions $\upsilon_\lambda(\square)$ depend on the "arm" and the "leg" of tile $\square$ in diagram $\lambda$. The arm is defined as all the tiles in a row to the right of tile $\square$, whereas the leg is defined as all the tiles in a column below tile $\square$. We say that the arms and the legs have positive parity $+$ when the arm or the leg is terminated by a complete tile, and negative parity $-$ when the terminating tile is a half-tile. For example, in Fig. 1 the leg has parity $-$, and the arm has parity $+$. Hook function $\upsilon_\lambda(\square)$ is defined according to the following table:

| Leg parity | Arm parity | $\upsilon_\lambda(\square)$ |
|:---:|:---:|:---:|
| even | even | $\dfrac{1-q^{2|\mathbf{leg}_\lambda(\square)|+2}t^{2|\mathbf{arm}_\lambda(\square)|}}{1-q^{2|\mathbf{leg}_\lambda(\square)|}t^{2|\mathbf{arm}_\lambda(\square)|+2}}$ |
| odd | even | $q^2\dfrac{1-q^{2|\mathbf{leg}_\lambda(\square)|}t^{2|\mathbf{arm}_\lambda(\square)|}}{1-q^{2|\mathbf{leg}_\lambda(\square)|}t^{2|\mathbf{arm}_\lambda(\square)|+2}}$ |
| even | odd | $\dfrac{1-q^{2|\mathbf{leg}_\lambda(\square)|+2}t^{2|\mathbf{arm}_\lambda(\square)|}}{1-q^{2|\mathbf{leg}_\lambda(\square)|}t^{2|\mathbf{arm}_\lambda(\square)|}}$ |
| odd | odd | $q^2$ |

$$\tag{44}$$

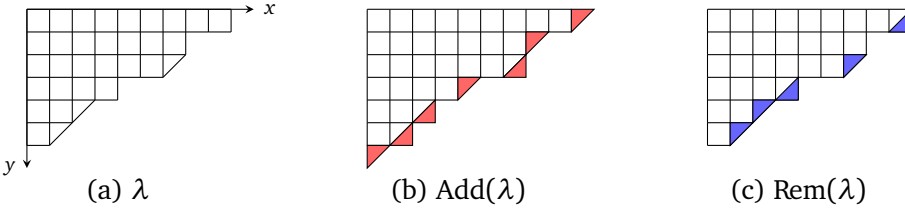

Figure 2: Sets Add($\star$) and Rem($\star$).

# 4   $\mathsf{T}(\widehat{\mathfrak{gl}}_{1|1})$ representation on super-Macdonald polynomials

In [2] we have shown that super-Jack polynomials $\mathsf{J}_\lambda$ (there these polynomials were named super-Schur polynomials) appear as a natural semi-Fock module of affine super-Yangian algebra $\mathsf{Y}(\widehat{\mathfrak{gl}}_{1|1})$. We were able to connect generators in the semi-Fock representation of this algebra to a super-Heisenberg algebra and the super-Young diagrams to molten crystals appearing in a realization of Yangian $\mathsf{Y}(\widehat{\mathfrak{gl}}_{1|1})$ as a BPS algebra of D-branes on a conifold. Here we established an independent combinatorial construction for $\mathsf{J}_\lambda$ in Sec. 3.2. We find this correspondence of two definitions surprising at the first glance and call it an *algebro-combinatorial duality*. It is natural to expect that Macdonald polynomials "refining" Jack polynomials would correspond under this duality to an algebra refining $\mathsf{Y}(\widehat{\mathfrak{gl}}_{1|1})$. The expected algebra [26,27,39](we denote it as $\mathsf{T}(\widehat{\mathfrak{gl}}_{1|1})$) is a close cousin of the Ding-Iohara-Miki (quantum toroidal) algebra and "refines" $\mathsf{Y}(\widehat{\mathfrak{gl}}_{1|1})$ in a loop direction. In this section we will construct a representation of $\mathsf{T}(\widehat{\mathfrak{gl}}_{1|1})$ on super-Macdonald polynomials $\mathsf{M}_\lambda$.

## 4.1   Moving in the partition space

A well-known and determining their relevance in the representation theory property of the Schur functions is that they are homomorphic to a $\mathfrak{gl}_\infty$ character ring, whose summation and multiplication reflect tensor summation and multiplication of representations. Then we end up with a clear and simple combinatorial rule how a product of two Schur functions is re-expanded in a linear combination of Schur functions in ring $\mathscr{R}$. In particular, a multiplication by the first Schur function $p_1$ acts on a Schur function of arbitrary diagram $\lambda$ in the following way: it is mapped with unit coefficients to all the Schur functions of new diagrams $\lambda'$ that differ from $\lambda$ by a single box. Analogously, a differentiation with respect to $p_1$ "removes" a box from all the possible places. Then it is reasonable to consider a basis in $\mathscr{R}$ spanned by Young diagrams, and operators of multiplication by $p_1$ and differentiation $d/dp_1$ as operators "adding" and "subtracting" boxes. This logic is naturally extended to Jack and Macdonald polynomials, although matrix coefficients for these operators are no longer units.

A similar picture we may observe in the super-case. Let us define two sets Add($\lambda$) and Rem($\lambda$) as sets of half-tiles $a$ that could be added or subtracted from a super-Young diagram $\lambda$ so that the new collection of tiles denoted as $\lambda \pm a$ respectively represents again a super-Young diagram. See an example in Fig. 2.

In this terms a multiplication by the elementary super-Macdonald polynomial $\mathsf{M}_{\{1/2\}} = p_{1/2}$ reads:

$$p_{1/2} \cdot \mathsf{M}_\lambda = \sum_{\triangledown \in \text{Add}(\lambda)} \mathbf{E}_{\lambda, \lambda+\triangledown} \mathsf{M}_{\lambda+\triangledown}. \tag{45}$$

Analogously, differentiation reads:

$$\frac{\partial}{\partial p_{1/2}} \mathsf{M}_\lambda = \sum_{\triangledown \in \text{Rem}(\lambda)} \mathbf{F}_{\lambda, \lambda-\triangledown} \mathsf{M}_{\lambda-\triangledown}. \tag{46}$$

In these two expressions matrix coefficients $\mathbf{E}_{\lambda,\lambda+\triangledown}$ and $\mathbf{F}_{\lambda,\lambda-\triangledown}$ are some functions of $q$ and $t$ we will compute in the next subsection. What is relevant in these two expressions is that multiplication by a super-Macdonald polynomial of diagram $\triangledown$ (resp. dual differentiation) adds (resp. removes) $\triangledown$ to all the vacant positions.

We could use the orthogonality property of the super-Macdonald polynomials to define diagram move amplitudes as:

$$\mathbf{E}_{\lambda,\lambda+\triangledown} = \frac{\left\langle \mathsf{M}_{\lambda+\triangledown}, p_{1/2} \cdot \mathsf{M}_\lambda \right\rangle_{\mathrm{M}}}{\langle \mathsf{M}_{\lambda+\triangledown}, \mathsf{M}_{\lambda+\triangledown} \rangle_{\mathrm{M}}}, \qquad \mathbf{F}_{\lambda,\lambda-\triangledown} = \frac{\left\langle \mathsf{M}_{\lambda-\triangledown}, \dfrac{\partial}{\partial p_{1/2}} \mathsf{M}_\lambda \right\rangle_{\mathrm{M}}}{\langle \mathsf{M}_{\lambda-\triangledown}, \mathsf{M}_{\lambda-\triangledown} \rangle_{\mathrm{M}}}. \tag{47}$$

Unfortunately, the other half-tile $\triangleleft$ is not an elementary super-Young diagram and we can not reinterpret a process of adding or removing it as an elementary action on the super-Macdonald polynomials. However we could consider a process of adding a complete tile $\square$ as a composite process of adding $\triangledown$ first, then $\triangleleft$, and construct respective amplitudes assuming that they split in this process. Let us note that for a complete tile $\square$ there is a respective elementary polynomial $\mathsf{M}_{\{1\}} = p_1$. So the following factorization is expected:

$$\langle \mathsf{M}_{\lambda+\square}, p_1 \cdot \mathsf{M}_\lambda \rangle_{\mathrm{M}} = \langle \mathsf{M}_{\lambda+\square}, \mathsf{M}_{\lambda+\square} \rangle_{\mathrm{M}} \times \mathbf{E}_{\lambda,\lambda+\triangledown} \times \mathbf{E}_{\lambda+\triangledown,(\lambda+\triangledown)+\triangleleft},$$
$$\left\langle \mathsf{M}_{\lambda-\square}, \frac{\partial}{\partial p_1} \mathsf{M}_\lambda \right\rangle_{\mathrm{M}} = \langle \mathsf{M}_{\lambda-\square}, \mathsf{M}_{\lambda-\square} \rangle_{\mathrm{M}} \times \mathbf{F}_{\lambda,\lambda-\triangledown} \times \mathbf{F}_{\lambda-\triangledown,(\lambda-\triangledown)-\triangleleft}. \tag{48}$$

We should warn the reader that the multiplication (differentiation) by $p_1$ process does not give only diagrams with a complete tile added (removed). As a byproduct there appear terms where half-tiles $\triangledown$ and $\triangleleft$ are moved separately. Relations (47) and (48) allow one to compute respective amplitudes explicitly.

## 4.2 A representation on super-Macdonald polynomials

Let us start this subsection with explicit expressions for the adding/removing a half-tile process amplitudes. To express these quantities we will need to define what are coordinates of tiles. We assume that coordinates $x_a$, $y_a$ of a half-tile $a = \triangledown, \triangleleft$ are coordinates of a complete tile $\square$ they belong to. And coordinates of complete tiles are defined from an integral mesh in the lower right quadrant assuming that axis $x$ is directed horizontally towards the right, and axis $y$ is directed vertically towards the bottom (see Fig. 2(a)), and the most upper left tile has coordinates $(0,0)$. Also we would like to introduce short-hand notations $x_{ab} := x_a - x_b$, $y_{ab} = y_a - y_b$.

In these terms amplitudes factorize:

$$\mathbf{E}_{\lambda,\lambda+a} = \Gamma_a(\lambda) \times \prod_{b\in\lambda} \Delta_{a,b}(x_{ab}, y_{ab}), \qquad \mathbf{F}_{\lambda,\lambda-a} = \Theta_a(\lambda) \times \prod_{b\in\lambda} \Xi_{a,b}(x_{ab}, y_{ab}), \tag{49}$$

where the products run over all half-tiles in diagrams, and element functions entering these expressions read:

$$\Gamma_\triangledown(\lambda) = \begin{cases} 1, & \text{if } x_\triangledown = 0, \\ \dfrac{1-t^2}{1-t^{2x_\triangledown}} (-1)^{\sum\limits_{i=1}^{x_\triangledown -1} \mathfrak{f}(\lambda_i)}, & \text{otherwise}, \end{cases} \tag{50a}$$
$$\Gamma_\triangleleft(\lambda) = \frac{1-q^{-2}t^{2x_\triangleleft+2}}{1-q^{-2}t^2} (-1)^{\sum\limits_{i=1}^{x_\triangleleft -1} \mathfrak{f}(\lambda_i)},$$

$$\Delta_{\triangledown,\triangledown}(x,y) = \Delta_{\vartriangle,\vartriangle}(x,y) = 1 \,,$$

$$\Delta_{\triangledown,\vartriangle}(x,y) = \begin{cases} 1\,, & \text{if } y > 0 \,, \\[2mm] \dfrac{1-q^2}{1-q^2 t^2}\,, & \text{if } (x,y) = (1,0)\,, \\[2mm] \dfrac{(1-t^{2x-2}q^{-2y+2})(1-t^{2x}q^{-2y})}{(1-t^{2x-2}q^{-2y})(1-t^{2x}q^{-2y+2})}\,, & \text{otherwise,} \end{cases} \tag{50b}$$

$$\Delta_{\vartriangle,\triangledown}(x,y) = \begin{cases} 1\,, & \text{if } y > 0 \text{ or } (x,y)=(0,0)\,, \\[2mm] \dfrac{(1-t^{2x}q^{-2y-2})(1-t^{2x+2}q^{-2y})}{(1-t^{2x+2}q^{-2y-2})(1-t^{2x}q^{-2y})}\,, & \text{otherwise,} \end{cases}$$

$$\Theta_{\triangledown}(\lambda) = \begin{cases} (-1)^{\sum\limits_{i=1}^{x_{\triangledown}-1} f(\lambda_i)}\,, & \text{if } y_{\triangledown}=0\,, \\[4mm] q^{2y_{\triangledown}}\dfrac{1-q^2}{1-q^{2y_{\triangledown}}}(-1)^{\sum\limits_{i=1}^{x_{\triangledown}-1} f(\lambda_i)}\,, & \text{otherwise,} \end{cases} \tag{50c}$$

$$\Theta_{\vartriangle}(\lambda) = q^{-2y_{\vartriangle}}\frac{1-q^{2y_{\vartriangle}}t^{-2}}{1-q^2 t^{-2}}(-1)^{\sum\limits_{i=1}^{x_{\vartriangle}-1} f(\lambda_i)}\,,$$

$$\Xi_{\triangledown,\triangledown}(x,y) = \Xi_{\vartriangle,\vartriangle}(x,y) = 1\,,$$

$$\Xi_{\triangledown,\vartriangle}(x,y) = \begin{cases} 1\,, & \text{if } x > 0\,, \\[2mm] \dfrac{1-t^2}{1-q^2 t^2}\,, & \text{if } (x,y)=(0,1)\,, \\[2mm] \dfrac{(1-t^{-2x+2}q^{2y-2})(1-t^{-2x}q^{2y})}{(1-t^{-2x+2}q^{2y})(1-t^{-2x}q^{2y-2})}\,, & \text{otherwise,} \end{cases} \tag{50d}$$

$$\Xi_{\vartriangle,\triangledown}(x,y) = \begin{cases} 1\,, & \text{if } x > 0 \text{ or } (x,y)=(0,0)\,, \\[2mm] \dfrac{(1-t^{-2x}q^{2y+2})(1-t^{-2x-2}q^{2y})}{(1-t^{-2x-2}q^{2y+2})(1-t^{-2x}q^{2y})}\,, & \text{otherwise.} \end{cases}$$

Using these expressions we construct operators acting on a family of super-Macdonald polynomials as a vector space. These operators add or remove a tile from the respective super-Young diagram with an amplitude we constructed earlier:

$$\hat{e}_0^+ \mathsf{M}_\lambda = \sum_{\triangledown \in \mathrm{Add}(\lambda)} \mathbf{E}_{\lambda,\lambda+\triangledown}\mathsf{M}_{\lambda+\triangledown}\,, \qquad \hat{e}_0^- \mathsf{M}_\lambda = \sum_{\vartriangle \in \mathrm{Add}(\lambda)} \mathbf{E}_{\lambda,\lambda+\vartriangle}\mathsf{M}_{\lambda+\vartriangle}\,,$$

$$\hat{f}_0^+ \mathsf{M}_\lambda = \sum_{\triangledown \in \mathrm{Rem}(\lambda)} \mathbf{F}_{\lambda,\lambda-\triangledown}\mathsf{M}_{\lambda-\triangledown}\,, \qquad \hat{f}_0^- \mathsf{M}_\lambda = \sum_{\vartriangle \in \mathrm{Rem}(\lambda)} \mathbf{F}_{\lambda,\lambda-\vartriangle}\mathsf{M}_{\lambda-\vartriangle}\,. \tag{51}$$

Then the actions of multiplication or differentiation by a time variable could be rewritten in terms of these operators in the following way:

$$p_{1/2}\cdot = \hat{e}_0^+\,, \qquad p_1\cdot = \left\{\hat{e}_0^+,\hat{e}_0^-\right\}\,,$$

$$\frac{\partial}{\partial p_{1/2}} = \hat{f}_0^+\,, \qquad \frac{\partial}{\partial p_1} = \left\{\hat{f}_0^+,\hat{f}_0^-\right\}\,. \tag{52}$$

At the moment we are not aware of generic expressions for all the time operators for the super-Macdonald polynomials since they become involved rather quickly. For the 3/2 time multiplication operator we suggest the following expression:

$$p_{3/2}\cdot = \frac{qt}{1-q^2}\frac{\left[\hat{e}_1^+,\{\hat{e}_{-1}^-,\hat{e}_0^+\}\right]-\left[\hat{e}_0^+,\{\hat{e}_{-1}^-,\hat{e}_1^+\}\right]}{2}\,, \tag{53}$$

whereas, for example, relations for ordinary Macdonald polynomials and $\mathsf{T}(\widehat{\mathfrak{gl}}_1)$ are expanded into infinite sums of raising operator modes (see [14, eq. (148)]), and a similar property is naturally expected from the first non-trivial even time $p_2$.

Nevertheless, for the cases of super-Jack (and super-Schur) polynomials analogous expressions in terms of $\mathsf{Y}(\widehat{\mathfrak{gl}}_{1|1})$ are well-known (cf. [2, eq. (21), (27)]):

$$
p_{1/2}\cdot = \hat{e}_0^+, \qquad p_1\cdot = \{\hat{e}_0^-, \hat{e}_0^+\}, \qquad p_j\cdot = \begin{cases} -\dfrac{\beta^{-1}}{j-1}\{\hat{e}_0^-, [\hat{e}_1^+, p_{j-1}\cdot]\}, & \text{if } \mathfrak{f}(j)=0, \\[2mm] -\dfrac{\beta^{-1}}{j-1/2}[\hat{e}_1^+, \{\hat{e}_0^-, p_{j-1}\cdot\}], & \text{if } \mathfrak{f}(j)=1, \end{cases}
$$

$$
\partial_{p_{1/2}} = \hat{f}_0^+, \qquad \partial_{p_1} = \{\hat{f}_0^-, \hat{f}_0^+\}, \qquad \partial_{p_j} = \begin{cases} \dfrac{1}{j}\{\hat{f}_0^-, [\hat{f}_1^+, \partial_{p_{j-1}}]\}, & \text{if } \mathfrak{f}(j)=0, \\[2mm] \dfrac{1}{j-1/2}[\hat{f}_1^+, \{\hat{f}_0^-, \partial_{p_{j-1}}\}], & \text{if } \mathfrak{f}(j)=1, \end{cases}
\tag{54}
$$

where generators $\hat{e}_k^\pm, \hat{f}_k^\pm$ for a representation of $\mathsf{Y}(\widehat{\mathfrak{gl}}_{1|1})$ on super-Jack polynomials:

$$
\hat{e}_k^+ \mathsf{J}_\lambda = \sum_{\triangledown \in \mathrm{Add}(\lambda)} \mathbf{E}_{\lambda, \lambda+\triangledown}(x_\triangledown - \beta y_\triangledown)^k \mathsf{J}_{\lambda+\triangledown}, \qquad \hat{e}_k^- \mathsf{J}_\lambda = \sum_{\vartriangle \in \mathrm{Add}(\lambda)} \mathbf{E}_{\lambda, \lambda+\vartriangle}(x_\vartriangle - \beta y_\vartriangle)^k \mathsf{J}_{\lambda+\vartriangle},
$$

$$
\hat{f}_k^+ \mathsf{J}_\lambda = \sum_{\triangledown \in \mathrm{Rem}(\lambda)} \mathbf{F}_{\lambda, \lambda-\triangledown}(x_\triangledown - \beta y_\triangledown)^k \mathsf{J}_{\lambda-\triangledown}, \qquad \hat{f}_k^- \mathsf{J}_\lambda = \sum_{\vartriangle \in \mathrm{Rem}(\lambda)} \mathbf{F}_{\lambda, \lambda-\vartriangle}(x_\vartriangle - \beta y_\vartriangle)^k \mathsf{J}_{\lambda-\vartriangle},
\tag{55}
$$

where amplitudes $\mathbf{E}_{\lambda, \lambda+a}, \mathbf{F}_{\lambda, \lambda-a}$ for adding and subtracting a tile could be derived easily from (49) in the limit $q = e^{\beta\epsilon}$, $t = e^\epsilon$, $\epsilon \to 0$.

## 4.3 Algebra $\mathsf{T}(\widehat{\mathfrak{gl}}_{1|1})$

To proceed let us introduce the following notations for a content of a half-tile in terms of its respective coordinates:

$$
\omega_\triangledown = t^{2x_\triangledown} q^{-2y_\triangledown}, \qquad \omega_\vartriangle = t^{2x_\vartriangle+1} q^{-2y_\vartriangle-1}.
\tag{56}
$$

It is natural to consider higher tile moving operators twisted with the $k^{\text{th}} \in \mathbb{Z}$ powers of tile contents:

$$
\hat{e}_k^+ \mathsf{M}_\lambda = \sum_{\triangledown \in \mathrm{Add}(\lambda)} \omega_\triangledown^k \mathbf{E}_{\lambda, \lambda+\triangledown} \mathsf{M}_{\lambda+\triangledown}, \qquad \hat{e}_k^- \mathsf{M}_\lambda = \sum_{\vartriangle \in \mathrm{Add}(\lambda)} \omega_\vartriangle^k \mathbf{E}_{\lambda, \lambda+\vartriangle} \mathsf{M}_{\lambda+\vartriangle},
$$

$$
\hat{f}_k^+ \mathsf{M}_\lambda = \sum_{\triangledown \in \mathrm{Rem}(\lambda)} \omega_\triangledown^k \mathbf{F}_{\lambda, \lambda-\triangledown} \mathsf{M}_{\lambda-\triangledown}, \qquad \hat{f}_k^- \mathsf{M}_\lambda = \sum_{\vartriangle \in \mathrm{Rem}(\lambda)} \omega_\vartriangle^k \mathbf{F}_{\lambda, \lambda-\vartriangle} \mathsf{M}_{\lambda-\vartriangle}.
\tag{57}
$$

Momentarily we will introduce a collection of Cartan operators $\psi_k^\pm, \chi_k^\pm, k \in \mathbb{Z}_{\geq 0}$ and will show that all together operators $e_j^\pm, f_j^\pm, j \in \mathbb{Z}$ and $\psi_k^\pm, \chi_k^\pm, k \in \mathbb{Z}_{\geq 0}$ form a representation of trigonometric deformation $\mathsf{T}(\widehat{\mathfrak{gl}}_{1|1})$ of the quiver Yangian algebra $\mathsf{Y}(\widehat{\mathfrak{gl}}_{1|1})$ [26,27] – a close supersymmetric cousin of the Ding-Iohara-Miki algebra [40–42] named also a quantum toroidal algebra in the literature [43–45].

To do so let us note that the adding/subtracting a tile amplitudes satisfy a set of relations we call *hysteresis* relations as they represent differences of amplitudes as one moves in the space of super-Young diagrams along closed loops:

$$
\frac{\mathbf{E}_{\lambda, \lambda+a}\mathbf{E}_{\lambda+a, \lambda+a+b}}{\mathbf{E}_{\lambda, \lambda+b}\mathbf{E}_{\lambda+b, \lambda+a+b}} \varphi_{ab}(\omega_a/\omega_b) = -1,
$$

$$
\frac{\mathbf{F}_{\lambda+a+b, \lambda+a}\mathbf{F}_{\lambda+a, \lambda}}{\mathbf{F}_{\lambda+a+b, \lambda+b}\mathbf{F}_{\lambda+b, \lambda}} \varphi_{ab}(\omega_a/\omega_b) = -1,
$$

$$
\mathbf{E}_{\lambda+a, \lambda+a+b}\mathbf{F}_{\lambda+a+b, \lambda+b} = -\mathbf{F}_{\lambda+a, \lambda}\mathbf{E}_{\lambda, \lambda+b},
\tag{58}
$$

$$
\mathbf{E}_{\lambda, \lambda+a}\mathbf{F}_{\lambda+a, \lambda} = \lim_{u \to 1}(1-u) \times \Psi_\lambda^{(a)}(u\,\omega_a),
$$

where

$$\Psi_\lambda^{(a)}(u) = \psi_0^{(a)}(u) \times \prod_{b \in \lambda} \varphi_{a,b}(u/\omega_b),$$

$$\psi_0^{(\triangledown)}(u) = \frac{1}{1-u}$$

$$\psi_0^{(\triangle)}(u) = \frac{1}{(1-q^{-2})(1-t^2)}\left(1-\frac{q}{t}u\right),$$

$$\varphi_{\triangledown,\triangledown}(u) = \varphi_{\triangle,\triangle}(u) = 1,$$

$$\varphi_{\triangledown,\triangle}(u) = \frac{\left(1-\frac{q}{t}u\right)\left(1-\frac{t}{q}u\right)}{\left(1-\frac{1}{qt}u\right)(1-qt\,u)},$$

$$\varphi_{\triangle,\triangledown}(u) = \frac{\left(1-\frac{1}{qt}u\right)(1-qt\,u)}{\left(1-\frac{q}{t}u\right)\left(1-\frac{t}{q}u\right)}. \tag{59}$$

It is useful to combine the raising and lowering operators into generating functions:

$$e^\pm(z) = \sum_k \hat{e}_k^\pm z^{-k}, \qquad f^\pm(z) = \sum_k \hat{f}_k^\pm z^{-k}. \tag{60}$$

We could use functions $\Psi_\lambda^{(a)}(u)$ to introduce new Cartan operators $\psi^\pm(z)$ and $\chi^\pm(z)$ on super-Macdonald polynomials via respective generating functions for them:

$$\psi^+(z)\,\mathsf{M}_\lambda := \left(\sum_{k=1}^\infty \hat{\psi}_k^+ z^{-k}\right)\mathsf{M}_\lambda = \left[\Psi_\lambda^{\triangledown}(z)\right]_\infty \times \mathsf{M}_\lambda,$$

$$\chi^+(z)\,\mathsf{M}_\lambda := \left(\sum_{k=0}^\infty \hat{\chi}_k^+ z^k\right)\mathsf{M}_\lambda = \left[\Psi_\lambda^{\triangledown}(z)\right]_0 \times \mathsf{M}_\lambda,$$

$$\psi^-(z)\,\mathsf{M}_\lambda := \left(\sum_{k=-1}^\infty \hat{\psi}_k^- z^{-k}\right)\mathsf{M}_\lambda = \left[\Psi_\lambda^{\triangle}(z)\right]_\infty \times \mathsf{M}_\lambda, \tag{61}$$

$$\chi^-(z)\,\mathsf{M}_\lambda := \left(\sum_{k=0}^\infty \hat{\chi}_k^- z^k\right)\mathsf{M}_\lambda = \left[\Psi_\lambda^{\triangle}(z)\right]_0 \times \mathsf{M}_\lambda,$$

where $[f(z)]_0$ and $[f(z)]_\infty$ are expansions of a rational function $f(z)$ into Taylor series around points $z = 0, \infty$ respectively.

The proposed generating functions satisfy $\mathsf{T}(\widehat{\mathfrak{gl}}_{1|1})$ relations ($a, b = \pm$):

$$\left[\psi^a(z), \psi^b(w)\right] = \left[\chi^a(z), \chi^b(w)\right] = \left[\psi^a(z), \chi^b(w)\right] = 0,$$

$$e^a(z)\,e^b(w) = -\varphi_{ab}(z/w)\,e^b(w)\,e^a(z),$$

$$f^a(z)\,f^b(w) = -\varphi_{ab}(z/w)^{-1}\,f^b(w)\,f^a(z),$$

$$\psi^a(z)\,e^b(w) = \varphi_{ab}(z/w)\,e^b(w)\,\psi^a(z),$$

$$\chi^a(z)\,e^b(w) = \varphi_{ab}(z/w)\,e^b(w)\,\chi^a(z), \tag{62}$$

$$\psi^a(z)\,f^b(w) = \varphi_{ab}(z/w)^{-1}\,f^b(w)\,\psi^a(z),$$

$$\chi^a(z)\,f^b(w) = \varphi_{ab}(z/w)^{-1}\,f^b(w)\,\chi^a(z),$$

$$\left\{e^a(z), f^b(w)\right\} = \delta_{ab} \times \left(\sum_{k \in \mathbb{Z}} z^k/w^k\right) \times (\psi^a(z) - \chi^a(z)).$$

We should draw reader's attention to the fact that we have derived a shifted representation of $\mathsf{T}(\widehat{\mathfrak{gl}}_{1|1})$ (see [39, 46–48] for details). This is reflected in uncompensated numbers of zeroes and poles in the vacuum charge functions $\psi_0^{(a)}(u)$ in (59) and a deviation of lower summation limits in (61) from zero. As another mild deviation from considerations in a QFT framework we could note a certain dis-balance of factors in (59) from a symmetry with respect to $u \to u^{-1}$. Naturally factors like $u^{\frac{1}{2}} - u^{-\frac{1}{2}} = 2\sinh(h/(2T)) \sim \prod_n (h - (2\pi n T))$, $u = e^{h/T}$, would appear as QFT answers and represent partition functions of simple scalar fields of mass/flavor fugacity $h$ on a thermal circle of circumference $2\pi T$. Non-chirality of a respective quiver description is responsible for the shifts and this dis-balance as the quiver arrow chirality dis-balance produces certain anomalies and non-zero beta-functions [49].

# 5 Towards super-Kostka amplitudes

## 5.1 Kostka numbers for even diagrams

Let us remind first the structure of upper-triangular decomposition (36) and its combinatorial meaning in the case when all the diagrams have purely even elements $\ell(\lambda^-) = \ell(\mu^-) = 0$ following [3].

Coefficients $K_{\lambda\mu}$ in this case have a transparent combinatorial meaning in terms of a summation over Young tableaux. In particular, for the Macdonald polynomials we have:

$$K_{\lambda\mu} = \sum_{\tau \in T(\lambda,\mu)} \prod_{i=1}^{\ell(\mu)} \tilde{B}_{\tau^i/\tau^{i-1}}, \tag{63}$$

where $T(\lambda, \mu)$ are Young tableaux represented in a form of a sequence of embedded diagrams $\tau^i$, and an "amplitude" of adding a vertical strip of length $|\nu/\sigma|$ to a diagram $\sigma$ so that the result is a diagram $\nu$ reads:

$$\tilde{B}_{\nu/\sigma} = \prod_{\square \in (C \setminus R)_{\nu/\sigma}} \frac{1 - q^{|\mathbf{leg}_\sigma(\square)|} t^{|\mathbf{arm}_\sigma(\square)|+1}}{1 - q^{|\mathbf{leg}_\sigma(\square)|+1} t^{|\mathbf{arm}_\sigma(\square)|}} \frac{1 - q^{|\mathbf{leg}_\nu(\square)|+1} t^{|\mathbf{arm}_\nu(\square)|}}{1 - q^{|\mathbf{leg}_\nu(\square)|} t^{|\mathbf{arm}_\nu(\square)|+1}}, \tag{64}$$

where $(C \setminus R)_{\nu/\sigma}$ is a set of all complete tiles in $\sigma$ belonging to the same columns but not the same rows as tiles in $\nu/\sigma$.

We should stress that expressions (63) are well defined for the both cases when $\lambda \geq \mu$ and $\lambda < \mu$, and even those cases when $\mu$ is not a Young diagram. However, for $\mu$ being a Young diagram and $\lambda < \mu$ set $T(\lambda, \mu) = \varnothing$, and, respectively, $K_{\lambda\mu} = 0$. This property ensures that decomposition (36) is upper-triangular indeed.

In the limit $q \to 1$, $t \to 1$ when the Macdonald polynomials reduce to the Schur polynomials these amplitudes $\tilde{B}_{\nu/\mu} \to 1$, and we arrive to the canonical result that in the case of the Schur functions coefficients $K_{\lambda\mu}$ count solely numbers of Young tableaux $T(\lambda, \mu)$ of shape $\lambda$ and weight $\mu$ without any extra weight factors. These numbers are called *Kostka numbers*. For example, at level 5 there are 7 purely even diagrams:

$$\young{}\; > \;\young{}\; > \;\young{}\; > \;\young{}\; > \;\young{}\; > \;\young{}\; > \;\young{} \tag{65}$$

In this case we have the following example expansion:

$$\mathsf{S}_{\young{}} = m_{\young{}} + m_{\young{}} + 2\,m_{\young{}} + 3\,m_{\young{}} + 5\,m_{\young{}}. \tag{66}$$

Here we reconstruct the Kostka numbers from the numbers of Young tableaux:

$$
T\left(\;\boxplus\;,\;\boxplus\;\right)=\left\{\;\begin{smallmatrix}1&2\\1&2\\1\end{smallmatrix}\;\right\},
$$

$$
T\left(\;\boxplus\;,\;\boxplus\!\square\;\right)=\left\{\;\begin{smallmatrix}1&2\\1&3\\1\end{smallmatrix}\;\right\},
$$

$$
T\left(\;\boxplus\;,\;\boxplus\!\!\boxminus\;\right)=\left\{\;\begin{smallmatrix}1&2\\1&3\\2\end{smallmatrix}\;,\;\begin{smallmatrix}1&2\\1&2\\3\end{smallmatrix}\;\right\}, \tag{67}
$$

$$
T\left(\;\boxplus\;,\;\boxplus\!\square\square\;\right)=\left\{\;\begin{smallmatrix}1&3\\1&4\\2\end{smallmatrix}\;,\;\begin{smallmatrix}1&2\\1&4\\3\end{smallmatrix}\;,\;\begin{smallmatrix}1&2\\1&3\\4\end{smallmatrix}\;\right\},
$$

$$
T\left(\;\boxplus\;,\;\square\square\square\square\;\right)=\left\{\;\begin{smallmatrix}1&4\\2&5\\3\end{smallmatrix}\;,\;\begin{smallmatrix}1&3\\2&5\\4\end{smallmatrix}\;,\;\begin{smallmatrix}1&3\\2&4\\5\end{smallmatrix}\;,\;\begin{smallmatrix}1&2\\3&5\\4\end{smallmatrix}\;,\;\begin{smallmatrix}1&2\\3&4\\5\end{smallmatrix}\;\right\}.
$$

In this section we will attempt to generalize the combinatorial structure of coefficients $K_{\lambda\mu}$ to the case of supersymmetric polynomials. We will call them *super-Kostka amplitudes* and will see that they are also representable now as summations over super-Young tableaux with certain weights. However, in what follows we will observe that supersymmetric analogs of $\tilde{B}_{\lambda/\mu}$ *do not factorize* in products of simple hook expressions in general.

## 5.2 Origin of tableaux

Following [3] first of all we would like to construct a basis of supersymmetric polynomials dual to the basis of $m_\lambda$ with respect to the appropriate norm.

It turns out we can do this in a universal way, therefore under the scalar product $\langle,\rangle$ we assume in this section any of (37), and under P we assume S, J or M respectively. For a half-integer number $x \in \mathbb{Z}/2$ let us define a column function in the following way:

$$
g_{\{x\}} := \begin{cases} \displaystyle\sum_{\lambda:\,|\lambda|=x,\,\ell(\lambda^-)=0} \frac{1}{\langle P_\lambda,P_\lambda\rangle}P_\lambda, & \text{if } \mathfrak{f}(x)=0, \\ \displaystyle\sum_{\lambda:\,|\lambda|=x,\,\ell(\lambda^-)=1} \frac{1}{\langle P_\lambda,P_\lambda\rangle}P_\lambda, & \text{if } \mathfrak{f}(x)=1, \end{cases} \tag{68}
$$

where the summations run over all the super-Young diagrams of appropriate size and having no or a single half-tile.

Then for a super-partition $\lambda$ we simply define:

$$
g_\lambda := \prod_i g_{\{\lambda_i\}}. \tag{69}
$$

Polynomials $g_\lambda$ are dual to $m_\lambda$ with respect to the corresponding norm used in (68):

$$
\langle g_\lambda, m_\mu\rangle = \delta_{\lambda\mu}. \tag{70}
$$

We could use this new basis in $\mathscr{S}$ to re-express the super-Kostka numbers as corresponding scalar products:

$$
K_{\lambda\mu} = \langle g_\mu, P_\lambda\rangle = \left\langle \prod_{x\in\mu} g_{\{x\}}, P_\lambda\right\rangle. \tag{71}
$$

It turns out that a multiplication on the right by a column function $g_{\{x\}}$ act in the space of polynomials also as an operator mapping diagram $\lambda$ to new diagrams $\nu$ *only* such that $\nu/\lambda$ is a vertical strip of length $x$:

$$
P_\lambda g_{\{x\}} = \sum_{\nu:\,|\nu/\mu|=x} B_{\nu/\lambda} P_\nu. \tag{72}
$$

For our definition of vertical strips applied to super-partitions see Sec. 2.2. We will discuss the structure of the column amplitudes $B_{\lambda/\mu}$ in the next subsection. At the moment let us rather draw immediate consequences of such an expansion.

Let us remind that the polynomial corresponding to a zero partition is given just by unity $P_\varnothing = 1$. Then we apply relation (72) consecutively:

$$
\begin{aligned}
g_{\{\mu_1\}} &= \sum_{\tau^1:\,|\tau^1/\varnothing|=\mu_1} B_{\tau^1/\varnothing}\, P_{\tau^1}\,, \\
g_{\{\mu_1\}}g_{\{\mu_2\}} &= \sum_{\tau^1:\,|\tau^1/\varnothing|=\mu_1}\;\sum_{\tau^2:\,|\tau^2/\tau^1|=\mu_2} B_{\tau^1/\varnothing}B_{\tau^2/\tau^1}\, P_{\tau^2}\,, \\
g_{\{\mu_1\}}g_{\{\mu_2\}}g_{\{\mu_3\}} &= \sum_{\tau^1:\,|\tau^1/\varnothing|=\mu_1}\;\sum_{\tau^2:\,|\tau^2/\tau^1|=\mu_2}\;\sum_{\tau^3:\,|\tau^3/\tau^2|=\mu_3} B_{\tau^1/\varnothing}B_{\tau^2/\tau^1}B_{\tau^3/\tau^2}\, P_{\tau^3}\,, \\
&\cdots
\end{aligned}
\tag{73}
$$

to arrive in the r.h.s. at sequences of diagrams $\varnothing \subset \tau^1 \subset \tau^2 \subset \ldots \subset \tau^{\ell(\mu)}$ forming all various super-Young tableaux of weight $\mu$.

Using the orthogonality property of the supersymmetric polynomials $P_\lambda$ for *super-Kostka* amplitudes we derive the following expression:

$$
K_{\lambda\mu} = \langle P_\lambda, P_\lambda \rangle \times \sum_{\{\tau^i\}\in T(\lambda,\mu)} \prod_{i=1}^{\ell(\mu)} B_{\tau^i/\tau^{i-1}}\,,
\tag{74}
$$

where $T(\lambda,\mu)$ is a set of all super-Young tableaux of shape $\lambda$ and weight $\mu$ (see Sec. 2.2). We could transform expression (74) to (63) if redefine:

$$
\tilde{B}_{\lambda/\mu} = \frac{\langle P_\lambda, P_\lambda \rangle}{\langle P_\mu, P_\mu \rangle} B_{\lambda/\mu}\,,
\tag{75}
$$

however here we find quantities $B_{\lambda/\mu}$ are more convenient to work with.

## 5.3 Miraculous form-factors in column amplitudes

Unfortunately, at the moment we are not aware of a generic expression for amplitudes $B_{\lambda/\mu}$ to complete expression (74) in the case of super-Macdonald polynomials except two simple cases when $g_{\{1/2\}} \sim p_{1/2}$ and $g_{\{1\}} \sim p_1$, as they are similar to operators (52). Some conjectural form for similar expressions could be found in [50, 51].

To illustrate rule (72) let us apply it to super-Schur functions for simplicity. In particular, we derive:

$$
S_{\{2,2,1/2\}} \cdot g_{\{5/2\}} = -\frac{7}{5} S_{\{7/2,2,3/2\}} - 3 S_{\{9/2,2,1/2\}} - \frac{5}{16} S_{\{3,2,3/2,1/2\}} - \frac{4}{3} S_{\{5/2,2,2,1/2\}} - \frac{3}{2} S_{\{7/2,2,1,1/2\}}\,,
\tag{76}
$$

where all the diagrams on the r.h.s. include the diagram on the l.h.s., so that the quotient is a vertical strip:

$$
\{7/2,2,3/2\} = \quad,\qquad \{9/2,2,1/2\} = \quad,\qquad \{3,2,3/2,1/2\} = \quad,
$$
$$
\{5/2,2,2,1/2\} = \quad,\qquad \{7/2,2,1,1/2\} = \quad.
\tag{77}
$$

On the other hand for super-Jack polynomials we are able to implement explicit expressions (54) for time-variables in terms of $Y(\widehat{\mathfrak{gl}}_{1|1})$ generators to rewrite the Pieri rules (72) for column multiplication in terms of tile adding amplitudes $\mathbf{E}_{\lambda,\lambda+\triangledown}$ and $\mathbf{E}_{\lambda,\lambda+\triangle}$. Let us do so for the next to trivial case:

$$\mathsf{J}_\lambda g_{\{1\}} = p_1 \mathsf{J}_\lambda = \left(\hat{e}_0^+ \hat{e}_0^- + \hat{e}_0^- \hat{e}_0^+\right) \mathsf{J}_\lambda = \sum_{\triangledown,\triangle} \left(\mathbf{E}_{\lambda,\lambda+\triangledown}\mathbf{E}_{\lambda+\triangledown,\lambda+\triangledown+\triangle} + \mathbf{E}_{\lambda,\lambda+\triangle}\mathbf{E}_{\lambda+\triangle,\lambda+\triangledown+\triangle}\right) \mathsf{J}_{\lambda+\triangledown+\triangle}, \quad (78)$$

where we sum over all possible insertions of two half-tiles into $\lambda$. The expression in brackets could be simplified further by applying hysteresis relations (58). To pass to super-Jack polynomials we simply take a limit $q = e^{\epsilon\beta}$, $t = e^\epsilon$, $u = e^{\epsilon z}$, $\epsilon \to 0$ in (59) and introduce useful notations:

$$u = x_\triangledown - \beta y_\triangledown, \qquad v = x_\triangle - \beta y_\triangle. \quad (79)$$

Then the expression in brackets factorizes nicely:

$$\begin{aligned}
B_{(\lambda+\triangledown+\triangle)/\lambda} &= \mathbf{E}_{\lambda,\lambda+\triangledown}\mathbf{E}_{\lambda+\triangledown,\lambda+\triangledown+\triangle} + \mathbf{E}_{\lambda,\lambda+\triangle}\mathbf{E}_{\lambda+\triangle,\lambda+\triangledown+\triangle} \\
&= \mathbf{E}_{\lambda,\lambda+\triangledown}\mathbf{E}_{\lambda+\triangledown,\lambda+\triangledown+\triangle} \times \left(1 - \varphi_{\triangledown,\triangle}\left(\frac{\omega_\triangledown}{\omega_\triangle}\right)\right) \\
&= \mathbf{E}_{\lambda,\lambda+\triangledown}\mathbf{E}_{\lambda+\triangledown,\lambda+\triangledown+\triangle} \times \left(1 - \frac{(u-v)(u-v+\beta-1)}{(u-v-1)(u-v+\beta)}\right) \\
&= -\frac{\beta\, \mathbf{E}_{\lambda,\lambda+\triangledown}\mathbf{E}_{\lambda+\triangledown,\lambda+\triangledown+\triangle}}{(u-v-1)(u-v+\beta)}.
\end{aligned} \quad (80)$$

We could repeat this procedure for $g_j$ to arrive to the following structure for $|\nu/\lambda| = j$:

$$B_{\nu/\lambda} = H_{\nu/\lambda}\Upsilon_j\left(u_1,\ldots,u_{\lceil j\rceil}; v_1,\ldots,v_{\lfloor j\rfloor}\right), \quad (81)$$

where $H_{\nu/\lambda}$ are simple completely factorizable expressions consisting of consequently applied amplitudes $\mathbf{E}_{\lambda,\lambda+a}$ as we add tiles to arrive from $\lambda$ to $\nu$ in some fixed order and some denominators from tile permutation factors $\varphi_{a,b}$. On the other hand, polynomial form-factors $\Upsilon_j$ arrive from passing to a common denominator and then combining various contributions and depend on coordinates $u_i$ and $v_i$ of upper $\triangledown$ and lower $\triangle$ half-tiles in $\nu/\lambda$. For instance,

$$\begin{aligned}
\Upsilon_{1/2}(u_1) &= 1, \\
\Upsilon_1(u_1; v_1) &= 1, \\
\Upsilon_{3/2}(u_1, u_2; v_1) &= 1 - w_{12} - w_{22}, \\
\Upsilon_2(u_1, u_2; v_1, v_2) &= 1 - w_{11} - w_{22} + w_{11}w_{22} + w_{12}w_{21},
\end{aligned} \quad (82)$$

where $w_{ab} = u_a - v_b$.

Form-factors $\Upsilon_j$ control the ultimate form of a vertical strip $\nu/\lambda$ as it is embedded into $\nu$. In particular, if we try to arrange tiles in $\nu$ in a such way that $\nu/\lambda$ is not a vertical strip the form-factor as a function of coordinates $u_i$, $v_i$ annihilates such an amplitude. For example, let us consider a vertical strip of length $3/2$. It contains two tiles $\triangledown$ and one tile $\triangle$. We could arrange them in a strip that is not vertical:

$$\boxed{\triangle\triangledown} \quad : \quad u_1 = c, \quad u_2 = c+1, \quad v_1 = c, \quad \text{then} \quad \Upsilon_{3/2}(c, c+1; c) = 0. \quad (83)$$

We observe that starting with level 2 form-factors (82) *do not factorize* in general and become rather involved quite rapidly. However, to observe this in practice one might need to consider diagrams of relatively high level for generic enough arrangement parameters $u_i$, $v_i$. For example,

$$\boxed{\phantom{xx}} \quad : \quad B_{\{7/2,5/2,2,1\}/\{3,2,3/2,1/2\}} = -\frac{\beta(12 + 18\beta + 7\beta^2)}{12(1+\beta)^4(2+\beta)(3+2\beta)}. \quad (84)$$

Then numerator in this expression does not factorize in a product of brackets $(c + d\beta)$ with $c, d \in \mathbb{Z}$, that is a consequence of the fact that $\Upsilon_2$ is a quadratic polynomial.

Nevertheless, we could always reduce our super-discussion to the ordinary symmetric functions by considering purely even diagrams $\ell(\lambda^-) = 0$ only. In this case we should consider only even levels $k \in \mathbb{Z}$ and arrange pairs of tiles $\triangledown$ and $\triangle$ in such a way that they form complete tiles $\square$ only. In this case arrangement parameters for two tiles in a pair are related $u_i = v_i$. Under these constraints form-factors factorize again (cf. factorized form in (64) and [3, chapter VI.6]):

$$\Upsilon_{k \in \mathbb{Z}}(u_1, \ldots, u_k; u_1, \ldots, u_k) = \prod_{1 \le i < j \le k} (u_i - u_j - 1)(u_i - u_j + 1) . \qquad (85)$$

In this factorized form it is easy to observe selection rules preventing complete tiles in $\nu/\lambda$ to gather in any arrangement other than a vertical strip. If an arrangement $\nu/\lambda$ is not a vertical strip then there are at least two complete tiles, say, the $i^{\text{th}}$ and the $j^{\text{th}}$ that stuck together in a row. For their coordinates we observe the following constraint $|u_i - u_j| = 1$ annihilating form-factor (85).

# 6 Conclusion

In this paper we developed a systematic combinatorial definition for constructed earlier super-symmetric polynomial families. These polynomial families generalize canonical Schur, Jack and Macdonald families so that the new polynomials depend on odd Grassmann variables as well. Members of these families are labeled by respective modifications of Young diagrams – super-Young diagrams. We showed that the super-Macdonald polynomials form a representation of a quantum toroidal (Ding-Iohara-Miki) super-algebra $\mathsf{T}(\widehat{\mathfrak{gl}}_{1|1})$. A supersymmetric modification for Young tableaux and Kostka numbers are also discussed.

As a conclusion we propose some more problems that might turn out to be of interest for further research:

- Schur polynomials are widely known for being a selected basis in the character ring of general Lie algebras $\mathfrak{gl}_\infty$. At the moment we are not aware if similar properties (except mentioned in the text orthogonality and upper-triangular decomposition) are inherited by super-Schur polynomials as well. Yet we hope that the super-Schur functions will become a source for new intriguing identities.

- As we have said in the introduction a favor paid to a start from a triangular decomposition can seem arbitrary. The fundamental principle inducing such a triangular decomposition is not fully clear to us. We know that it is a kind of basic in quasi-classics – the triangular property of the decomposition matrices is reminiscent of Stokes transition matrices where triangularity is induced by a gradient flow irreversibility. However, the way it is lifted to the full-fledged quantum level is somewhat mysterious. An alternative (mirror dual) structure to gradient flows is an exceptional collection in a triangulated category [52]. We hope that (super-)Kostka amplitudes are representable as certain parallel transports with some Berry connection [53–56] on some QFT moduli spaces. In this case triangularity might acquire a physical explanation.

- In this text we have constructed only lower level operators $p_{1/2}\cdot$, $p_1\cdot$, $\partial_{p_{1/2}}$, $\partial_{p_1}$ in terms of generators of $\mathsf{T}(\widehat{\mathfrak{gl}}_{1|1})$. It would be intriguing to derive generic expressions for all $p_j$, $j \in \mathbb{Z}/2$.

- As we were able to re-express differential operators in super-times $p_j$ in terms of $\mathsf{T}(\widehat{\mathfrak{gl}}_{1|1})$ there should be an inverse transform. In particular, Cartan operators $\hat{\psi}_k^\pm$, $\hat{\chi}_k^\pm$ should form Hamiltonians of an integrable system where super-Macdonald polynomials are families of eigen functions.

- We derived that column multiplication amplitudes $B_{\lambda/\mu}$ in general are proportional to form-factor functions $\Upsilon_{|\lambda/\mu|}$. Properties of these form-factors are rather intriguing: despite they do not factorize in general, they are "aware" of the shape of $\lambda/\mu$, so that if $\lambda/\mu$ is not a vertical strip $\Upsilon_{|\lambda/\mu|} = 0$. It might be interesting to pursue studying properties of these functions further and develop ultimately a systematic construction for all $\Upsilon_j$. Determinantal expressions for functions analogous to $\Upsilon_j$ were proposed in [50, 51].

- In addition to the standard limit $q = e^{\beta\epsilon}$, $t = e^\epsilon$, $\epsilon \to 0$ when Macdonald polynomials converge to Jack polynomials there is another well-defined limit $q = \omega_r e^{\beta\epsilon}$, $t = \omega_r e^\epsilon$, $\epsilon \to 0$, where $\omega_r$ is the $r^{\text{th}}$ root of unity. In the latter case Macdonald polynomials give rise to Uglov polynomials that form Fock representations of affine Yangians $\mathsf{Y}(\widehat{\mathfrak{gl}}_r)$ [14, 15, 57]. We hope that in the case of supersymmetric polynomials this limit will be also well-defined and produce representations of $\mathsf{Y}(\widehat{\mathfrak{gl}}_{r|r})$ or other affine super-Yangians.

## Acknowledgments

**Funding information**    The work was partially funded within the state assignment of the Institute for Information Transmission Problems of RAS. Our work is partly supported by grant RFBR 21-51-46010 ST_a (D.G., A.M., N.T.), by the grants of the Foundation for the Advancement of Theoretical Physics and Mathematics "BASIS" (A.M. and N.T.).

## A    Hamiltonians

Here we would like to work out some hints on the inverse map to (52) incorporating expressions for Hamiltonians as differential operators in super-times $p_j$.

First let us remind expressions for lower level Hamiltonians for ordinary Macdonald polynomials emerging for purely even diagrams $\ell(\lambda^-) = 0$:

$$
H_+^{(0)} \mathsf{M}_\lambda = \left( \sum_{\square \in \lambda} t^{2x_\square} q^{-2y_\square} \right) \times \mathsf{M}_\lambda \,,
$$
$$
H_-^{(0)} \mathsf{M}_\lambda = \left( \sum_{\square \in \lambda} t^{-2x_\square} q^{2y_\square} \right) \times \mathsf{M}_\lambda \,.
\tag{A.1}
$$

Expressions for them may be derived in a closed form of contour integrals around $z = 0$ in a spectral parameter plane:

$$
H_+^{(0)} = \frac{1}{(1-t^2)(q^{-2}-1)} \left[ \oint \frac{dz}{z} \exp\left( \sum_{k=1}^\infty \frac{1-t^{2k}}{k} p_k z^k \right) \exp\left( \sum_{k=1}^\infty (q^{-2k}-1) \frac{\partial}{\partial p_k} z^{-k} \right) - 1 \right],
$$
$$
H_-^{(0)} = \frac{1}{(1-t^{-2})(q^2-1)} \left[ \oint \frac{dz}{z} \exp\left( \sum_{k=1}^\infty \frac{1-t^{-2k}}{k} p_k z^k \right) \exp\left( \sum_{k=1}^\infty (q^{2k}-1) \frac{\partial}{\partial p_k} z^{-k} \right) - 1 \right].
\tag{A.2}
$$

For the case of super-Macdonald polynomials we expect four lower level Hamiltonians corresponding to lower modes of $\psi^+(z)$, $\psi^-(z)$, $\chi^+(z)$, $\chi^-(z)$ with the following eigen values:

$$H_+^{(1)} \mathsf{M}_\lambda = \left( \sum_{\triangledown \in \lambda} t^{2x_\triangledown} q^{-2y_\triangledown} \right) \times \mathsf{M}_\lambda \,, \qquad H_-^{(1)} \mathsf{M}_\lambda = \left( \sum_{\triangledown \in \lambda} t^{-2x_\triangledown} q^{2y_\triangledown} \right) \times \mathsf{M}_\lambda \,,$$

$$H_+^{(2)} \mathsf{M}_\lambda = \left( \sum_{\triangle \in \lambda} t^{2x_\triangle} q^{-2y_\triangle} \right) \times \mathsf{M}_\lambda \,, \qquad H_-^{(2)} \mathsf{M}_\lambda = \left( \sum_{\triangle \in \lambda} t^{-2x_\triangle} q^{2y_\triangle} \right) \times \mathsf{M}_\lambda \,. \tag{A.3}$$

Let us introduce the following notations:

$$\mathbf{P}_\lambda := \frac{\prod_{x \in \lambda^+} (1 - t^{2x})}{z_\lambda} \prod_i p_{\lambda_i} \,, \qquad \mathbf{D}_\lambda := \frac{\prod_{x \in \lambda^+} (q^{-2x} - 1)x}{z_\lambda} \prod_i \frac{\partial}{\partial p_{\lambda_i}} \,. \tag{A.4}$$

In these terms for the first Macdonald Hamiltonian we derive the following expansion:

$$H_+^{(0)} = \sum_{\lambda : \ell(\lambda^-)=0} \sum_{\mu : \ell(\mu^-)=0} \delta_{|\lambda|,|\mu|} \mathbf{P}_\lambda \mathbf{D}_\mu \,. \tag{A.5}$$

We will need the following notation:

$$\xi_k = \frac{1 - q^{2k} t^{2k}}{q^{2k-2}} \,. \tag{A.6}$$

In these terms we have for two of four super-Hamiltonians:

$$H_+^{(1,2)} = H_+^{(0)} + \frac{1}{1 - q^2 t^2} \sum_{\lambda : \ell(\lambda^-)=1} \sum_{\mu : \ell(\mu^-)=1} \delta_{|\lambda|,|\mu|} Z^{(1,2)}(\lambda, \mu) \mathbf{P}_\lambda \mathbf{D}_\mu \,, \tag{A.7}$$

where $Z^{(1,2)}(\lambda, \mu) = Z^{(1,2)}(\mu, \lambda)$ and for non-zero elements we have the following expressions. For the first Hamiltonian we have:

- Level $^1/_2$: $Z^{(1)}(\{^1/_2\}, \{^1/_2\}) = \xi_1$.

- Level $^3/_2$: $Z^{(1)}(\{^3/_2\}, \{^3/_2\}) = \xi_2$, $Z^{(1)}(\{^3/_2\}, \{1, ^1/_2\}) = \xi_1$.

- Level $^5/_2$: $Z^{(1)}(\{^5/_2\}, \{^5/_2\}) = \xi_3$, $Z^{(1)}(\{^5/_2\}, \{2, ^1/_2\}) = \xi_1$, $Z^{(1)}(\{^5/_2\}, \{^3/_2, 1\}) = \xi_2$, $Z^{(1)}(\{^5/_2\}, \{1, 1, ^1/_2\}) = \xi_1$, $Z^{(1)}(\{^3/_2, 1\}, \{^3/_2, 1\}) = \xi_1$.

- Level $^7/_2$: $Z^{(1)}(\{^7/_2\}, \{^7/_2\}) = \xi_4$, $Z^{(1)}(\{^7/_2\}, \{3, ^1/_2\}) = \xi_1$, $Z^{(1)}(\{^7/_2\}, \{^5/_2, 1\}) = \xi_3$, $Z^{(1)}(\{^7/_2\}, \{2, ^3/_2\}) = \xi_2$, $Z^{(1)}(\{^7/_2\}, \{2, 1, ^1/_2\}) = \xi_1$, $Z^{(1)}(\{^7/_2\}, \{^3/_2, 1, 1\}) = \xi_2$, $Z^{(1)}(\{^7/_2\}, \{1, 1, 1, ^1/_2\}) = \xi_1$, $Z^{(1)}(\{^5/_2, 1\}, \{^5/_2, 1\}) = \xi_2$, $Z^{(1)}(\{^5/_2, 1\}, \{2, ^3/_2\}) = \xi_1$, $Z^{(1)}(\{^5/_2, 1\}, \{^3/_2, 1, 1\}) = \xi_1$.

- ...

For the second Hamiltonian we have:

- Level $^3/_2$: $Z^{(2)}(\{^3/_2\}, \{^3/_2\}) = \xi_1$.

- Level $^5/_2$: $Z^{(2)}(\{^5/_2\}, \{^5/_2\}) = \xi_2$, $Z^{(2)}(\{^5/_2\}, \{^3/_2, 1\}) = \xi_1$.

- Level $^7/_2$: $Z^{(2)}(\{^7/_2\}, \{^7/_2\}) = \xi_3$, $Z^{(2)}(\{^7/_2\}, \{^5/_2, 1\}) = \xi_2$, $Z^{(2)}(\{^7/_2\}, \{2, ^3/_2\}) = \xi_1$, $Z^{(2)}(\{^7/_2\}, \{^3/_2, 1, 1\}) = \xi_1$, $Z^{(2)}(\{^5/_2, 1\}, \{^5/_2, 1\}) = \xi_1$.

- ...

Lifting of these formulas to a generating function like (A.2) will be described elsewhere.

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
