# Peer review of "Supersymmetric polynomials and algebro-combinatorial duality"

_SciPost Physics, doi:SciPost Phys. 17, 119 (2024)_

## Round 1 · Referee Report · Anonymous (Referee 1) · 2024-8-18

Strengths

1- The work has very strong originality. In particular, a systematic way is presented to construct the super-Macdonald polynomials and Kostka numbers. 2- The algebra T(\hat{gl}_{1|1}) introduced in this article opens up potential applications of the super symmetric polynomials to the string theory and supersymmetric gauge theories.

Weaknesses

1- The article only covers the mathmatical aspects of the super -Macdonald polynomials, and its connection with physics such as the integrable models is to be explored in the future.

Report

The authors presented a systematic way to construct super-Schur, super-Jack, and super-Macdonald symmetric polynomials based on super Young diagrams and the attached Grassmann variables. A trigonometric version of the affine super Yangian is also introduced in the paper and the authors showed that the super Macdonald polynomials give a module of the trigonometric algebra. The article is well-written and organized, and the tools developed in the paper are expected to be applicable in the future study of representation theory and supersymmetric gauge theories. For this reason, I would like to recommend the manuscript be published after clarifying the following minor points.

Requested changes

  1. The canonical Schur polynomials (and the skew ones) are deeply related to the representation theory of A-type Lie algebras (and Yangian algebras). Do the super-Schur polynomials also give a certain kind of character to Lie superalgebras/super Yangians?

  2. Is there a simple way to see the relation between the trigonometric algebra T(\hat{gl}{1|1}) and the Lie superalgebra gl?}? Is it possible to further generalize the algebra to gl_{m|n

  3. The affine super-Yangian algebra Y(\hat{gl}_{1|1}) is expected to give a supersymmetric Calogero-Sutherland-like model with the super-Jack polynomials being the eigenfunctions. There have been many efforts in the literature to construct supersymmetric extensions of such a model and its eigenbasis, e.g. arXiv:1205.0784. Are the super-Jack polynomials built in this article a completely new version or do they match with some known results in the literature?

Recommendation

Ask for minor revision

---

## Round 1 · Referee Report · Anonymous (Referee 2) · 2024-8-19

Report

This paper generalizes the canonical Schur, Jack and Macdonald polynomials to their supersymmetric counterparts, with the help of Grassmann variables and newly-defined super-partitions etc.. Especially, with the so-called “algebro-combinatorial duality” introduced, the super-Macdonald polynomials serve as an representation of a DIM super-algebra, the refined version of affine super-Yangian algebra.

It is an important progress relating quantum algebra, which is worthy of more attention. I recommend publication of this nice work.

Just one quick remark: the authors may explain the reason why the sequence inequalities should be strict for odd numbers, below eq. (2.2).

Recommendation

Publish (easily meets expectations and criteria for this Journal; among top 50%)

---

## Round 2 · Referee Report · Anonymous (Referee 2) · 2024-9-25

Report

The updated version is recommended for publication.

Recommendation

Publish (easily meets expectations and criteria for this Journal; among top 50%)

---

## Round 2 · Referee Report · Anonymous (Referee 1) · 2024-9-25

Report

I recommend to publish the revised version of the manuscript.

Recommendation

Publish (easily meets expectations and criteria for this Journal; among top 50%)

---

## Round 2 · Author Response

Dear Editor-in-charge and Referees,

First of all we would like to thank the Referees for interesting questions.
Concerning the question of the first referee of the strictness for inequalities (2.2) for half-integer numbers. The reason is that we would treat half-integer numbers as fermions in what follows, so there could not be repetitions of those. However we wanted to keep this section as containing mostly bare definitions one could refer to, and to reveal their meaning in what follows.

All the best, Dmitry.

---

## Round 2 · List of Changes

Below are replies for the requests for the minor corrections of the other referee (enumeration coincides with referee's list): 1. Unfortunately, at the moment we are not aware if the super-Schur polynomials represent characters of anything constructive. We added this remark to further problems in concluding Section 6. 2. Absolutely, both quantum toroidal algebras and Yangians contain the algebra they are built on, usually as generators e_k, f_k, ... of lower mode number k. However, usually, for the quantum toroidal algebras this should not be directly the algebra itself rather its quantum version U_q(g). However we did not aim to describe T(\hat gl_{1|1}) from scratch in this text, rather we refer to the definition. We added a more explicit reference to the sources to the preamble of Section 4. 3. Indeed, our construction produces super-Jack and super-Macdonald polynomials similar to the other constructions from the literature. We added more new relevant references [5-11] including the reference the referee mentioned.

Other modifications in v2: 1. We fixed some typos. 2. We fixed some of author affiliations.

---

## Editorial Decision

published